# A model of long-term population growth with an application to Central West Argentina

**Jacob Freeman**[1,2☉]*, **Adolfo F. Gil**[3,4☉], **Eva A. Peralta**[3☉], **Fernando Franchetti**[3☉], **José Manuel López**[5☉], **Gustavo Neme**[3☉]

**1** Anthropology Program, Utah State University, Logan, UT, United States of America, **2** The Ecology Center, Utah State University, Logan, UT, United States of America, **3** Instituto de Evolución, Ecología Histórica y Ambiente, Consejo Nacional de Investigaciones Científicas y Técnicas (IDEVEA, CONICET & UTN). J. J. Urquiza 314, San Rafael, Mendoza, Argentina, **4** Facultad de Filosofía y Letras, Universidad Nacional de Cuyo, Mendoza, Argentina, **5** Instituto Argentino de Investigaciones de las Zonas Áridas (IADIZA, CCT CONICET Mendoza), Mendoza, Argentina

☉ All these authors are contributed equally to this work.
* jacob.freeman@usu.edu

**Data Availability Statement:** The data are associated with the paper are deposited on github and published via Zenodo. The data are freely available at this DOI: https://doi.org/10.5281/zenodo.11390810.

## Abstract

We propose an Ideal Specialization Model to help explain the diversity of population growth trajectories exhibited across archaeological regions over thousands of years. The model provides a general set of expectations useful for guiding empirical research, and we provide a concrete example by conducting a preliminary evaluation of three expectations in Central West Argentina. We use kernel density estimates of archaeological radiocarbon, estimates of paleoclimate, and human bone stable isotopes from archaeological remains to evaluate three expectations drawn from the model's dynamics. Based on our results, we suggest that innovations in the production of food and social organization drove demographic transitions and population expansion in the region. The consistency of population expansion in the region positively associates with changes in diet and, potentially, innovations in settlement and social integration.

## Introduction

Over the last three decades, archaeologists have made significant advances documenting the long-term population ecology of human societies [1–20]. Key empirical findings include:

(1) Human populations sometimes display periods of long-term, exponential-like growth during the Holocene [9, 10, 12, 16, 21–23]. This exponential-like expansion occurs on nearly every continent, regardless of subsistence base [10, 16, 22–25]. However, exponential-like growth was disrupted by periods of population recession and, in some cases, populations oscillating for centuries to a few millennia around a relatively constant size [2, 4, 7, 10–12, 21, 26–28].

(2) Human populations experienced an agricultural demographic transition–the ADT [20]. This transition empirically describes an interval of population growth associated with the adoption of agriculture among many different archaeological regions. Most researchers agree that the population growth was partly generated by the innovation of food production that

**Funding:** This research was supported by the Agencia Nacional de Promoción de la Investigación, el Desarrollo Tecnológico y la Innovación [PICT 2019–04447; PICT 2020-0684, PICT 2021–I-A-00891, and PICT 2022-04-00016], the Consejo Nacional de Investigaciones Científicas y Técnicas, and the Universidad Tecnológica Nacional. We are also thankful for support from the Past Global Changes (PAGES) project, which in turn received support from the Swiss Academy of Sciences and the Chinese Academy of Sciences.

**Competing interests:** NO authors have competing interests.

raised a region's carrying capacity, changing the biocultural environment to favor either increased fertility [19], decreased age-independent mortality [3], or both, and, thus, an initial gain in net reproduction. However, population vital rates (fertility and mortality) must have moved back toward equilibrium over a few centuries [19, 20, 29]. Although originally conceived as a singular event, many researchers document multiple waves of an ADT [7, 29, 30] and even document multiple waves of demographic transitions among human populations, regardless of subsistence base, during the Holocene [1, 2, 4].

The above patterns raise three fundamental questions. (1) What mechanisms enable and constrain the long-term (thousands of years), exponential-like expansion of human populations? (2) Why do some regions display more inter-generational stability, including differing amplitudes of potential population cycles, than other regions? (3) Is the ADT a revolutionary event, or is the ADT one manifestation of a more general process in which innovations in the production of resources sometimes generate demographic transitions among human societies? Answering these questions requires a two pronged research effort. First, the development of models useful for generating hypotheses about the diversity of population growth trajectories observed among archaeological regions [1, 2, 13], and, second, a library of case studies that integrate multiple lines of evidence useful for evaluating hypotheses about the drivers of long-term population growth. Here, we contribute to this effort by building on and studying an Ideal Specialization Model [1]. We use the model to generate expectations about the diversity of long-term trajectories of population expansion observed in Central West Argentina (CWA), where people adopted agriculture to various degrees over the last 3,000 years. We conduct a preliminary evaluation of these expectations given the evidence currently available, and we discuss how the model and its application to CWA generates research questions that can guide future data collection. Finally, we speculate on the factors that may generate the diversity of population growth trajectories displayed among archaeological regions during the Holocene.

## The ideal specialization model

The Ideal Specialization Model builds on a modified version of the logistic model [31] to investigate how the interaction of climate and innovations may generate a diversity of long-term trajectories of human population growth and decline [1]. Formally, we write the change in population as,

$$\dot{p}(t) = rp(t) - \frac{Sp(t)^2}{K}. \tag{1}$$

Where $\dot{p}$ is the change in human population; $r$ is the maximum rate of population gain (1/time); $S$ is the cost of social integration (1/time), $0 < S \leq 1$, and $K$ is the constraint on the productivity of resources used by humans in a given area. The term $rp$ captures the tendency of a population to increase exponentially. The term $-\frac{Sp(t)^2}{K}$ captures the dampening effect of competition for resources on population growth. This baseline model separates out social integration, $S$ and biophysical constraints, $K$ on the productivity of resources. This separation acknowledges that both social and biophysical processes may constrain the availability of resources.

Eq (1) is applicable to any species of social animal, but two additional processes are needed to describe changes in human populations over thousands of years. (1) Changes in climate that impact biophysical constraints on the productivity of resources [13, 32], and (2) a race between innovations that raise the productivity of resources and biophysical constraints on the productivity of resources [1, 4, 9, 33–36].

To add process (1) to the model, we write: $K_i(t) = C + \Delta(t)$. This states that biophysical constraints on resource productivity, $K_i(t)$, change over time as a function of the mean climate controlled productivity of a region, $C$ and stochastic variability, $\Delta(t)$ around the long-run mean climate. This proposition follows previous work proposing that the climate controlled productivity of ecosystems impacts the equilibrium population density of humans [13, 32], and the well documented relationship between environmental productivity and human population density in the ethnographic record at a global scale [37–39].

To add process (2), we model the impact of innovations that increase the productivity of resources, temporarily releasing biophysical constraints on $K_i(t)$. A large class of formal models, which elsewhere have been called Malthus-Boserup models, attempt to describe the race between innovation and biophysical limits on the productivity of resources [4, 9, 33–36, 40–42]. Following this previous work, we assume that populations have access to a set of latent innovations generated internally through experimentation or externally through cultural transmission. Given this background diversity of potential innovations, whether populations adopt the innovations or not depends on their ability to recognize and respond to signals of population pressure [9, 33, 34, 36, 43]. Population pressure refers to the fitness of an average individual relative to the minimum tolerable fitness of an average individual in a social-ecological setting [1, 34, 36], which we call $f_{min}$. Whenever the mean fitness of a population is less than $f_{min}$, populations should receive signals of instability in their food production system, and this instability may provide an incentive to adopt innovations for producing food and/or socially controlling access to key resources.

Dividing both sides of Eq (1) by $p$ gives the mean *per capita* fitness–which is equivalent to the mean *per capita* population growth rate ($\dot{p}/p$) of the system,

$$f \equiv \dot{p}/p = r - \left(\frac{Sp(t)}{K}\right). \tag{2}$$

Where per capita fitness ($f$) decreases linearly as the population of an area increases. The y-intercept of this linear function denotes the maximum rate of population gain ($r$), and the slope of the curve is controlled by the ratio of the cost of social integration to resource constraints ($\frac{Sp(t)}{K}$). The equilibrium population density, or maximum population pressure, occurs when net fitness equals 0, on average, and a population can just replace itself.

Given Eq (2), we describe the effect of the adoption of innovations on biophysical constraints as: $K_i(t) = A_i - B_i p(t - d)$. This equation states that biophysical constraints are a function of the ability of an infrastructure system, $A_i$ to augment the climate controlled productivity of resources in a region, and the effect of human population density on the productivity of resources, $B_i p(t - d)$ for a given infrastructure system, $i$, and delay time $d$.

For instance, as noted above, climate impacts the population density of ethnographically recorded societies. However, holding climate equal, differences in technology and social organization also impact human population densities [9, 37, 44]. Thus, specific forms of infrastructure, such as irrigation based agriculture, can augment the productivity of a given climate and result in higher population density. We introduce a delay, $d$ because oscillations in animal populations often occur due to delays between consumer and resource interactions in ecosystems [45]. As discussed by [1], the delay function provides a general framework to capture the effect of a lag between harvest pressure on resources and ecosystem response to such harvest pressure. For example, if a population primarily hunts animals for food in an area, it will take time for an initial population occupying the area to learn all of the best places to hunt and begin to deplete an animal population. This process is captured by the classic logistic model. The delay used here simulates the effect of additional behavioral processes that occur in

response to increased population density. For instance, animals change their ranges in response to harvest pressure and, thus, dampen the effect of human harvest on reproduction. Further, if hunting returns decline, humans increase the tempo of their movement, rotating hunting grounds more frequently, temporarily relieving pressure on animal populations. Together, both processes can create a delay in the manifestation of the impact of increasing population density on the surplus of production generated through hunting.

Putting the climate and innovation processes together, we write the change in resource constraints as,

$$K_i(t) = A_i - B_i p(t - d) + (C + \Delta(t)). \tag{3}$$

The change in resource constraints depends upon the infrastructure adopted by humans, the effect of the infrastructure system on the productivity of resources as human population increases in a fixed area, and climate.

Finally, an 'if, then, else' function captures the process of innovations potentially induced by population pressure. If $f_{min} < f$, then $K_1(t)$, else $K_2(t)$. For simplicity, we study two functions, $K_1(t)$ and $K_2(t)$. This allows us to describe the demographic transition dynamics of the model driven by an innovation in infrastructure system ($A_1$ to $A_2$), coupled with climate ($C + \Delta(t)$). Importantly, the transition from $K_1(t)$ to $K_2(t)$ captures an 'easy' innovation in productive capacity. This is to say that a set of latent innovations exists to modify the infrastructure of a production system. The lack of a $K_3$ denotes a soft innovation ceiling (i.e., a situation in which a set of latent innovations has become very small).

## Model dynamics

The model suggests three main insights: (1) Population expansion over thousands of years must be generated by a continuous release of biophysical constraints on the production of resources and/or the costs of social integration to access resources [1]. (2) Delays in the impact of humans on a resource base create instability in a population system. Specifically, in the presence of an innovation ceiling, more productive climates or infrastructure systems lead to more violent damped oscillation (i.e., recessions) of the population system. (3) The ability of a population to detect signals of population pressure is as critical as the number of latent innovations available that might release constraints on the productivity of resources. If signals of population pressure are obscure, in the sense that individuals cannot detect or react to declines in fitness until *per capita* fitness reaches zero, then climate variability is a necessary mechanism to trigger innovations and, consequently, demographic transitions.

To illustrate the above insights, we present the results of three sets of numerical experiments. In the first set of experiments, we study a simple version of the ISM to clearly define the mechanisms of long-term population expansion. First, we simplify the ISM by treating mean climate as the only parameter that determines $K_i(t)$ (i.e., innovation is impossible and population density has no effect on $K_i(t)$). Second, we hold climate constant, relax the assumption of an innovation ceiling, and assume that $A_i$ is the only parameter that impacts $K_i(t)$. In the second and third sets of experiments, we progressively increase the complexity of the model to understand how the presence of an innovation ceiling, the feedback of human population density on the abundance of resources, $K_i(t)$, and short-term climate variability impact population dynamics.

**Set #1: Population expansion.** Fig 1A and 1B illustrate two potential mechanism behind long-term population expansion. First, Fig 1A illustrates an experiment in which there are no latent innovations, individuals cannot detect population pressure ($f_{min} = 0$), and the long-term climate driven productivity of the system changes slowly. The point where each curve crosses

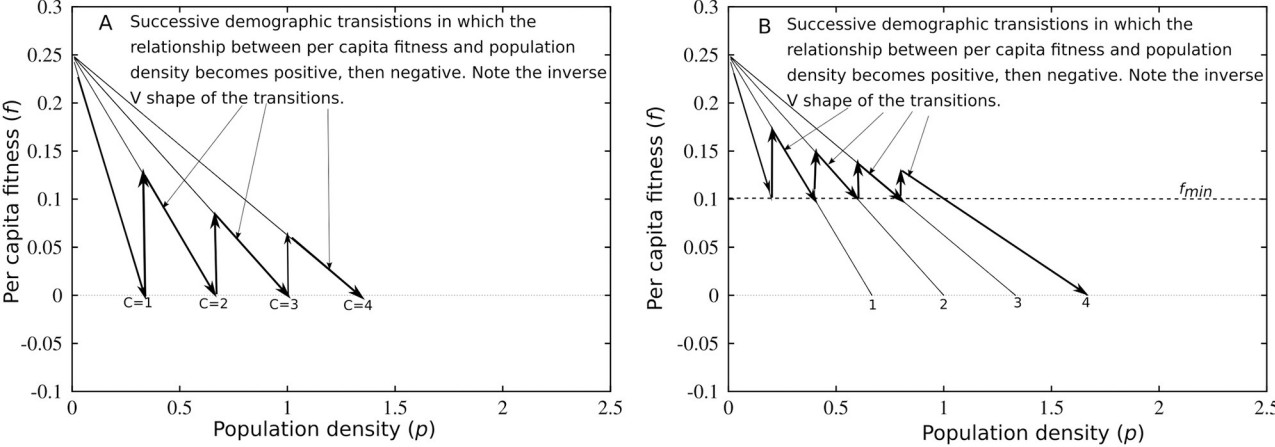

**Fig 1. A-Population density on the x-axis and *per capita* fitness on the y-axis.** The four black curves illustrate the relationship between population density and *per capita* fitness at four different levels of climate driven resource productivity, *C*. The arrows indicate how a population may change over time, with demographic transitions occurring only if climate change increases the productivity of resources ($r = 0.25$, $S = 0$, $A_1 - A_2 = 0$, $B_i = 0$, $d = 0$) B-Population density on the x-axis and *per capita* fitness on the y-axis. The four black curves illustrate the relationship between population density and *per capita* fitness for successively more productive infrastructure systems ($r = 0.25$, $S = 0$, $B_i = 0$, $d = 0$, $C = 1$).

the x-axis is an equilibrium where the population does not change. The arrows indicate how the population system changes over time. For instance, in an environment where the long-term mean climate suitability equals 1 ($C = 1$), an initial population entering the environment will experience a *per capita* fitness of approximately 0.25. Over time, the population grows and *per capita* fitness declines. The decline in *per capita* fitness occurs until the system reaches equilibrium at the point marked $C = 1$. In the absence of a directional change in the mean climate, the population would simply remain at equilibrium. However, suppose that over 3,000 years the productivity of ecosystems, on average, doubles every thousand years due to positive climate changes. The arrow moving up from the $C = 1$ point to the $C = 2$ curve and so on until the $C = 4$ point illustrates the population dynamics. Populations would go through successive demographic transitions, with *per capita* fitness increasing each time the climate improves. After each climate improvement, the population moves toward equilibrium, with *per capita* fitness only increasing again with the next pulse of improved climate. At a macro-demographic scale, the inverse V (or U in noisy systems) relationship between population density and *per capita* fitness signals a demographic transition.

Second, Fig 1B illustrates a thought experiment in which innovations can occur and $f_{min}$ = 0.1. This means that individuals experience signals of population pressure whenever *per capita* fitness crosses the 0.1 threshold, and, in response, they can adopt innovations that increase the productivity of resources. In this scenario, each curve experiences a demographic transition driven by the adoption of more productive infrastructure for producing food. Again, following the curve labeled "1" (shaped like a lightning bolt), as a population begins to grow, *per capita* fitness declines. When *per capita* fitness falls below 0.1, an innovation occurs to infrastructure system $A_2$, the productivity of resources increases, and, consequently, *per capita* fitness increases. Then *per capita* fitness declines again. If there are no more available latent innovations, the population will decline to equilibrium at the point labeled "1". However, if further innovation can occur, then the population will transition to the curve "2", again, experiencing a demographic transition. These waves of growth will continue until latent innovations become constrained, in this case when the population enters the equilibrium labeled "4".

Unlike the climate improvement pathway above, the population system does not reach an equilibrium, unless the system approaches an innovation ceiling. The ISM assumes that declines in fitness stimulate individuals to adopt latent innovations that raise the productivity of an environment. The notion of latent innovations proposes an "ideal sequence" in which some innovations in infrastructure (tools, landscape modifications, transport, etc.) have a higher upfront fitness cost than others. However, once the infrastructure is in place, the population experiences a temporary increase in mean *per capita* fitness due to increases in the productivity of resources. For example, in many regions, including CWA, populations often specialize on domesticated plants slowly over time [7, 46]. In fitness terms, initially adopting domesticated plants has a low fitness cost of entry, while adopting intensive agriculture has a high cost of entry. Thus, even though adopting intensive agriculture may be more productive, there is a sequence of first minimal adoption, driving a demographic transition, and then, as fitness declines, an adoption of more intensive agriculture that generates another demographic transition. This is a population and innovation ratchet that can result in waves of demographic transitions and long-term population expansion [2, 29, 30].

In summary, the ISM describes climate and infrastructure as substitutes. As a consequence, there are four possible pathways toward long-term population expansion. (1) Infrastructure does not change, but climate improves, on average, over thousands of years. (2) Climate does not change, but repeated innovations in infrastructure drive population expansion over thousands of years. (3) Infrastructure deteriorates, but improvements in climate are bigger than the negative impacts of infrastructure deterioration. (4) Flip (3) around: Climate change is negative, but the gains from infrastructure innovations outweigh the productivity losses from climate, and, thus, population expands despite declines in climate suitability.

**Set #2: Delays, climate, and damped oscillations.** Fig 2 illustrates the population dynamics of the model for two mental experiments designed to understand how climate and innovation impact population stability. In the first experiment, we study the effect of increases in the productivity of the infrastructure system $A_2$ relative to $A_1$ (i.e., $A = A_2 - A_1$) in the presence of a delayed impact of humans on resources and an innovation ceiling (i.e., no potential $A_3$). Fig 2A and 2B illustrate that as the distance between $A_2$ and $A_1$ increases, a population experiences a larger, longer demographic transition, and a larger recession as the population approaches equilibrium. The recession is denoted by the hooked shape of the curves in Fig 2A. The larger hooks indicate longer periods of time with negative *per capita* fitness. Fig 2B illustrates the expansion-recession pattern over time. Note the much larger humped shape of the population curve at $A = 3$. The expansion and recession into equilibrium occurs because the combination of a larger increase in productivity and the delay weakens the balancing feedback of competition on population growth. Thus, growth races ahead of competition until the resource constraint suddenly occurs and *per capita* fitness turns negative. This occurs until the system enters equilibrium [1].

In the second experiment, we ask how changing the long-term, mean suitability of the climate impacts population stability. Fig 2C and 2D replicate panels A and B, except that now we also hold $A_2 - A_1 = 0.5$ and vary $C$. Fig 2C illustrates that as the mean climate becomes more suitable, the size of the demographic transition declines (the inverse V shape becomes smaller), equilibrium population increases, and, a more suitable climate leads to a larger overshoot and population recession. The size of the demographic transition declines simply because innovations that generate a constant magnitude of increase in productivity have less of a positive impact on *per capita* fitness in more suitable climates. The recession dynamic is illustrated over time in Fig 2D by the sudden uptick in population growth around 19 generations and the increasingly humped shape of the curve as $C$ increases. Again, the overshoot occurs because the more suitable climate, in combination with the delay and innovation in infrastructure,

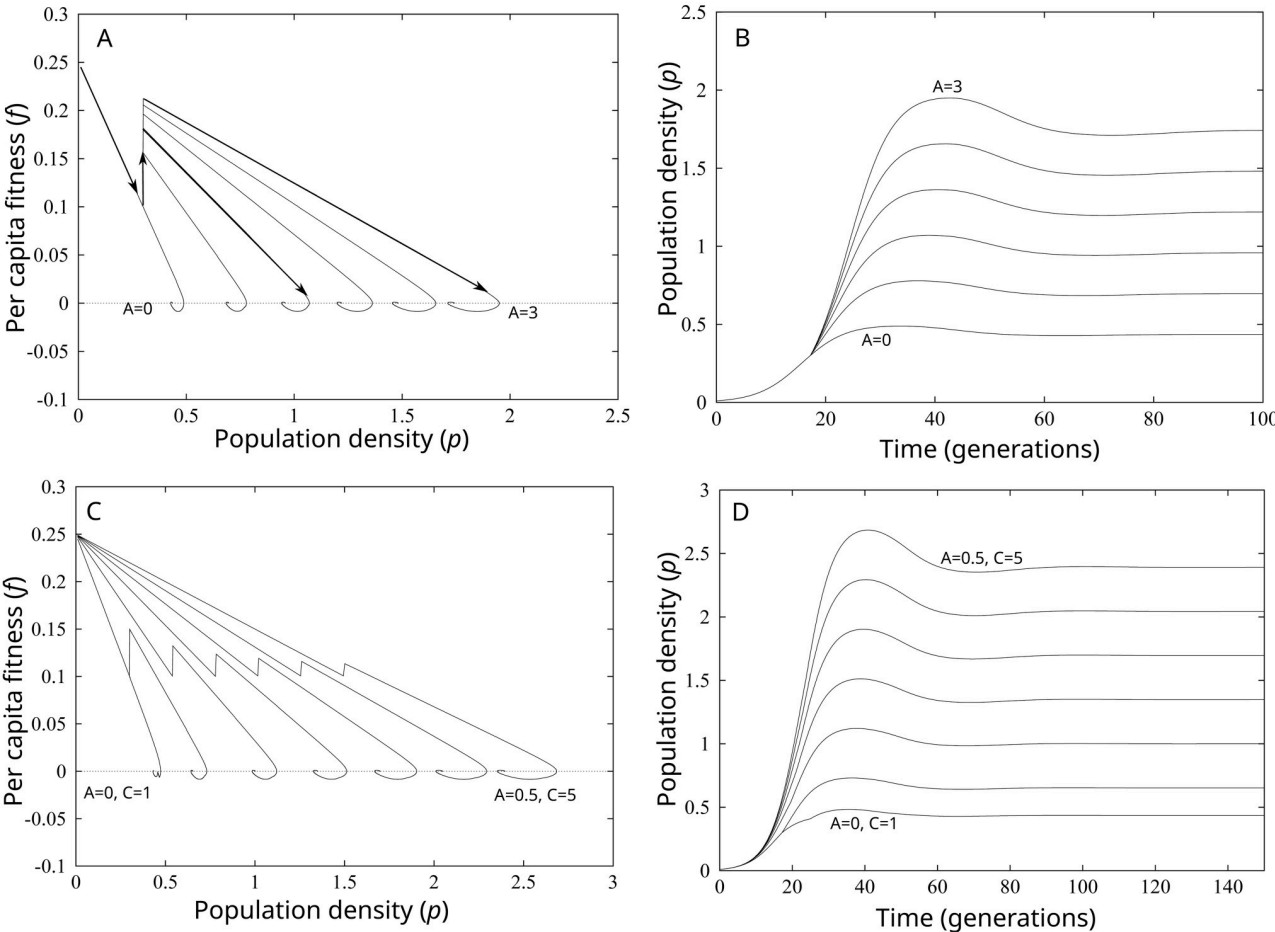

**Fig 2. A-Demographic transitions in a population system beginning at $A = 0$. All other parameters held constant: $S = 0.5$, $d = 25$, $C = 1$, $B_2 = -0.3$, and $r = 0.25$.** The delayed impact of human populations on the resource results in an overshoot and recession. B-Population dynamics over time with a delay. As $A = A_2 - A_1$ increases, the overshoot and recession increases in length and intensity. C-Demographic transitions in a population system beginning at $A = 0$, $C = 1$. In each successive curve, we hold $A = 0.5$, $S = 0.5$, $d = 25$, $B_2 = -0.3$, and $r = 0.25$. As $C$ increases, equilibrium population size also increases, the size of the demographic transition decreases, and the overshoot and recession intensifies. D-Population dynamics over time. As $C$ increases, the size of the overshoot and recession increases.

weakens the effect of competition for resources. Consequently, growth races ahead of competition until resource constraints suddenly become relevant. At that point, competition overpowers growth, population declines, and the system eventually enters equilibrium.

**Set #3: Detecting population pressure and short-term climate variability.** Fig 3 illustrates the effects of interactions between changes in the ability of a population to detect population pressure, mean climate suitability, and shorter-term climate variability on population dynamics. The dashed green curves in Fig 3A illustrate the effect of lowering $f_{min}$ to 0 in a setting with no short-term variability around the long-term climate mean. Lowering $f_{min}$ simulates a constraint on the ability of an average individual to detect population pressure. In this setting, the population experiences no demographic transition because it enters equilibrium before an innovation in infrastructure can occur. In Fig 3A and 3B, these equilibria are labeled $C_1$ and $C_3$, respectively. Once in equilibrium, the population enters a Malthusian trap in which, on average, individuals can just replace themselves. The simple point is that either an innovation ceiling (few latent infrastructure changes available) or, here, a constraint on the

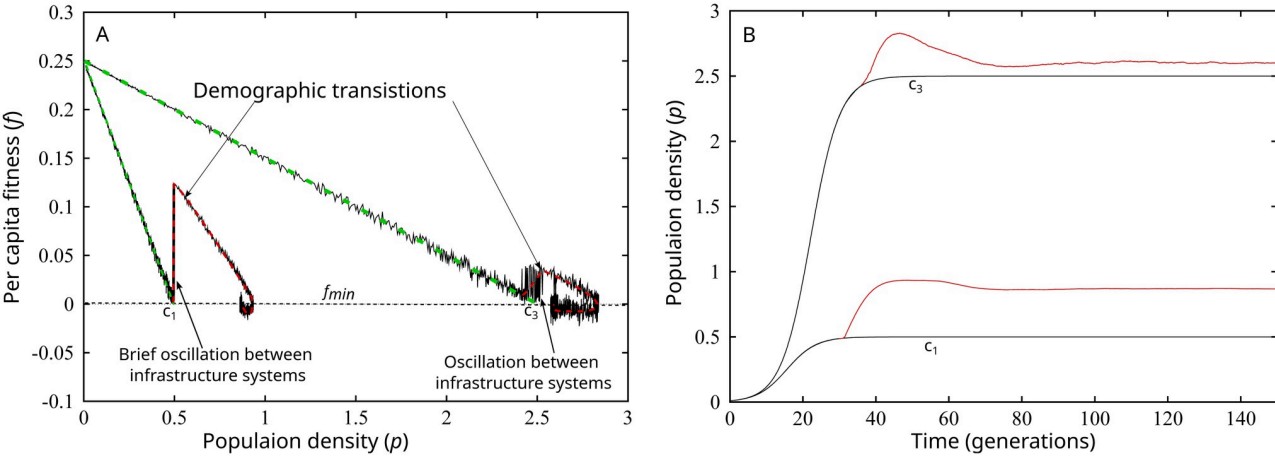

**Fig 3. A–Population density the x-axis and *per capita* fitness on the y-axis.** The black curves illustrate the relationship between population density and *per capita* fitness when $C = 1$ and $C = 3$. All other parameters are held equal at $A = 0.5$, $S = 0.5$, $d = 25$, $B_2 = -0.3$, $r = 0.25$, and *Delta* $= 0.01$. The green dashed curve illustrates what would occur in an environment with a constant mean climate. The red dashed curve illustrates the trend of the stochastic black curve when a demographic transition occurs. B–Change in population over time in stationary and stochastic climates of $C = 1$ and $C = 3$. The red curves illustrate the demographic transitions induced by climate variability highlighted in Fig 3A.

ability of a population to detect population pressure snuffs out a population ratchet. As noted above, improvements in the mean climate over thousands of years could generate a long-term population expansion in this scenario. However, there is another pathway to population expansion.

The black and red curves in Fig 3A illustrate the effect of adding short-term climate variability to the system. For example, beginning at a population of near 0 on the $C_1$ curve, population grows toward equilibrium at 0.5. The black curve illustrates how *per capita* fitness varies as population increases. Unlike the stable climate, however, the population does not enter equilibrium at $C_1$. Rather, for a short period of time, the system transitions between the $A_1$ and $A_2$ infrastructure systems, until a demographic transition occurs. When the demographic transition occurs, the average population grows and displays, as above, the recession into equilibrium. Again, the demographic transition is larger in the less suitable overall mean climate, and the recession is larger in the more suitable climate. These patterns are displayed over time in Fig 3B. The more suitable mean climate displays a much smaller demographic transition but a more humped shaped recession into equilibrium. In short, climate variability is a potential mechanism that generates a wider search of the potential state space of the population system. Such variability may induce innovations in production, even if a population has difficulty detecting population pressure.

## An example application to CWA

We conduct a preliminary analysis of population expansion in CWA to provide a concrete example of how the ISM can guide empirical research, help formulate contrasting hypotheses, and generate future directions for research. In general, CWA is arid with limited primary productivity [47, 48]. Rainfall is higher to the west in the foothills of the Andes and lower to the east in the lowland deserts due to the rain-shadow of the Andes [48, 49]. Further, researchers in CWA document that the region was sparsely inhabited until end of the Mid-Holocene. However, after 4,000 years ago, the archaeological record displays a stronger signal of human occupation, including sites in previously unoccupied areas [24, 50]. After 3,500 cal BP, three

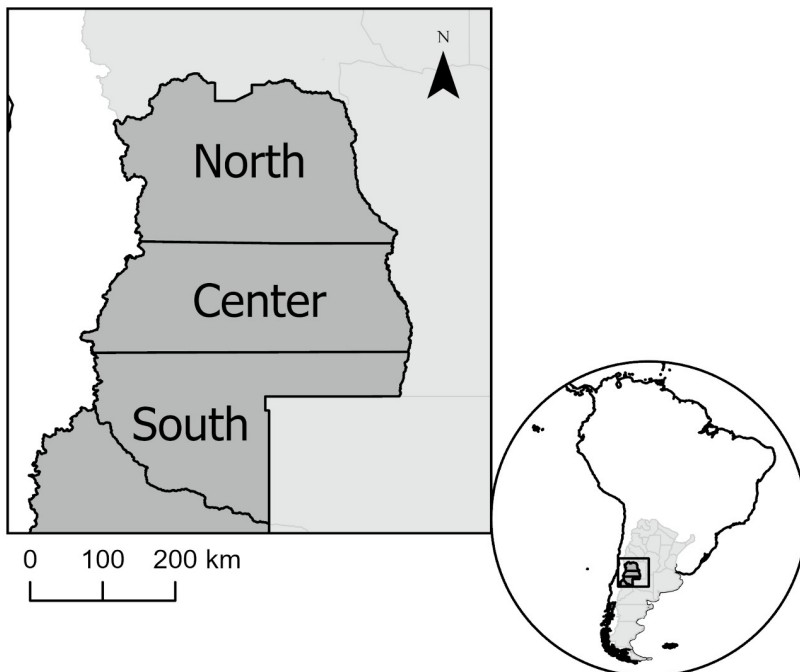

**Fig 4. Map of the archaeological regions discussed in the text.** Vector base map data available via CC BY license from Natural Earth (public domain at http://www.naturalearthdata.com) and The Humanitarian Data Exchange (public domain at https://data.humdata.org/dataset/cod-ab-arg). Figure produced using ArcGIS Pro 3.2 software.

areas of CWA (the Northern, Central, and Southern areas in Fig 4) varied in terms of the subsistence and social infrastructures that people created to produce and consume resources. Thus, CWA provides a region in which the mean productivity was very low and similar across three areas with divergent histories of changes in food production.

In the historical record, subsistence adaptations varied along the North-South gradient displayed in Fig 4. In the North, sedentary or semi-sedentary Huarpe horticulturalists occupied the area down to the Rio Diamante river valley [51–53]. In the Central and Southern areas, Puelche and Pehuenche hunter-gatherers occupied the areas from the Diamante south to the province of Neuquén, about 35–40 degrees south latitude [52, 53]. Early archaeological research projected this cultural-geographic pattern back 2,500 years, associating its origin with the arrival of maize agriculture in the region [54, 55]. Recent work provides a more nuanced picture [7, 56–59].

Table 1 summarizes the changes in subsistence, production, and settlement technologies in the three areas of CWA and differences between the areas in four temporal phases. Phase 1 (3,700–2,401 cal BP) documents the first consistent use of the region by hunter-gatherers following the intermittent occupation recorded during the Mid-Holocene [24, 50]. Phase 2 (2,400–1,301) documents the spread of agriculture into CWA and, traditionally, the development of mixed foraging and farming adaptations [6, 7, 55, 57, 60]. Phase 3 (1,300–501 cal BP) documents the period of time traditionally interpreted as the highest population density in the region with larger trade networks and more evidence of substantial settlements, particularly in the Northern area [7]. Finally, Phase 4 (500–300 cal BP) documents the final prehistoric and the beginning of disruptions generated by Inka [61] and Spanish intrusion [59, 62].

Table 1 helps illustrate two main points. First, all three areas display evidence of declines in the importance of hunting larger, more valued game and more evidence of seed use, especially

**Table 1. Summary of changes in subsistence and technology by four phases of time in the Northern, Central, and Southern areas of CWA.**

| Phase | Subsistence | Production Technology | Settlement Technology |
|---|---|---|---|
| **Northern Area** | | | |
| 3.7K-2.4K (1) | Consume camelids, *Neltuma*, *Geofroea*, and *Schinus*. Artiodactyl index over 70%, first presence of domesticates (*Chenopodium quinoa* and *Lagenaria siceraria*) after 3K cal BP | Projectile technology focus on hunting big/medium size fauna, use of atlatl, milling stones | Hearths, little evidence of structures |
| 2.39K-1.3K (2) | Considerable use of domesticates like *Cucurbita*, *Zea mays*, and *Chenopodium quinoa*. Earliest direct date on maize ca. 2,000 cal BP. Artiodactyl index 50% | Start to use bow and arrow, initial ceramic technology, milling stones increase in frequency | Hearths, pits, little evidence of structures |
| 1.29K-0.5K (3) | Higher dependence on domesticates, probably incorporation of domestic animals, Artiodactyl index less than 30% | Stylistic ceramic differentiation, milling stones increase | Hearths, pit-house structures by 1,000 cal BP, pits, effective occupation at higher altitudes |
| 0.5K (4) | Contraction of farming and herding to mountain areas. Re-advance of hunter gatherer way of life in the lowlands | Stylistic ceramic technology, Viluco ceramic style, milling stones | High variability in settlement, Inka occupation, first Hispanic settlements |
| **Central Area** | | | |
| 3.7K-2.4K (1) | Dependence on camelid hunting, plants like *Neltuma*, *Geofroea*, and *Schinus*. Artiodactyl index 80% | Atlatl and dart technology, milling stones | Hearths. No evidence of structures or storage pits |
| 2.39K-1.3K (2) | First evidence of domesticates (*Zea mays*, *Cucurbita*, *Phaseolus*, *Lagenaria*, *Chenopodium quinoa*), earliest direct date on maize ca. 2,000 cal BP. Artiodactyl index falls to an average of 40%. | Milling stones and mortars. Start to use bow and arrow, ceramics present, but less frequent than in the North; | Hearths. No evidence of structures or storage pits |
| 1.29K-0.5K (3) | Domesticates present. Wild resource plants reach their maximum diversity, with the exception of the mountains, Artiodactyl index 20% | Milling stones and mortars, stylistic ceramic present but less common than the North, potentiallly fishing technology | Hearths. High elevation effective occupation, no evidence of structures or storage pits |
| 0.5K (4) | Fewer sites with *Zea mays*, *Phaseolus*, and *Cucurbita*, Artiodactyl index reaches 70% | Mortars and milling stones, projectile technology focuses on hunting medium size fauna | Hearths, no evidence of structures or storage pits |
| **Southern Area** | | | |
| 3.7K-2.4K (1) | Hunter gatherers focus on camelid hunting, armadillos, and rhea eggs shells, Artiodactyl index 70%, few plant remains | Use of atlatl and dart, small milling stones | Hearths, no evidence of structures or storage pits |
| 2.39K-1.3K (2) | Camelids main resource, Artiodactyl index 60%, smaller wild plant species enter to the diet (*Maihuenia*, *Hofmansegia*, *Ephedra* | Projectile technology focus on hunting big/medium size fauna, bow and arrow, first plain ceramics, milling stones | Hearths, no evidence of structures or storage pits |
| 1.29K-0.5K (3) | Increase in wild plant diversity, low return rate plants added to the diet (*Suaeda*, *Berberis*, *Chenopodium papulosum*). Artiodactyl index 40% | High frequency of deep milling stones, ceramics, projectile technology focuses on small/medium size fauna | Hearths, large milling stones, no evidence of structures or storage pits |
| 0.5K (4) | Artiodactyl index 50%, less diversity of wild plant taxa | Deep milling stones, projectile technology focuses on small/medium size fauna | Hearths, no evidence of structures or storage pits |

domesticates in the Northern and Central areas, over time. For example, the Artiodactyl index estimates the ratio of larger, higher valued animals to high value plus small, lower valued animals in an archaeological assemblage. Across all three areas, this index decreases from Phase 1 to Phase 3. Coincident with these declines, more evidence exists for the use of seeds as food resources, including more evidence of milling stones and the use of at least some ceramics in all three regions by Phase 3.

Second, between areas we observe key differences in the incorporation of innovations in food production into the regional economies. The Late Holocene archaeological record of the Northern area indicates that domesticates were adopted earliest, around 3,000 cal BP in Phase 1, and then maize was adopted at the beginning of Phase 2. Subsequently, the Northern area displays evidence for maize-based agriculturalists during Phases 2 and 3 [7, 54, 60, 61, 63] that we do not observe in the Central or Southern areas. This includes many more maize macrofossils [64, 65], more substantial architecture and storage pits [63, 66], more substantial cemeteries [67], and larger quantities of decorated and simple ceramics from a range of vessel forms

and sizes [68]. During Phases 2 and 3, the archaeological record of the Central area displays clear evidence of maize consumption and macrofossils [55, 59]. However, little evidence for investment in residential structures, storage, and few ceramics exist in the area [56, 69]. In the Southern area, hunter-gatherers persisted throughout the Holocene, though with evidence of increasing seed production after 2,000 cal BP in Phase 2 and, especially, Phase 3 [70–73].

The ISM suggests three expectations for the long-term expansion of population in CWA.

1. We should observe longer periods of population expansion in the Northern area and shorter population expansions in the Central and Southern areas. This is because the archaeological record of the North, which includes the emergence storage facilities, ceramics, and larger settlements, suggests multiple innovations in food production and social integration that should have successively ratcheted equilibrium population density higher and higher. However, in the Southern and Central areas, population expansion should be shorter due to less evidence of innovations in food production, especially in maize production and social integration over time. The corollary of this expectation is that the Southern and Central regions should spend more time near demographic equilibrium.

2. If population expansion were driven by innovations in food production, then evidence for increasing population density should associate with evidence of more investment in the production of energy dense resources. In the areas where populations adopted maize, long-term increases in population density should associate with the adoption of maize and other domesticates, as well as specialization on the production of domesticates. In the Southern area, increases in population density should associate with changes in the production of wild resources, especially an increased use of wild plants, if such changes partly drive long-term population expansions.

3. If changes in the mean suitability of climate drove population expansion, then we should observe that improvements in climate suitability associate with population expansion. Further, we should not observe larger increases in population density than we would expect due to climate change alone during periods of population density expansion in all three areas. This expectation follows from the ISM's assumption that climate suitability and food production infrastructure act as substitutes that impact resource constraints in an area. Thus, if we account for long-term changes in population density driven by changes in climate, any remaining variation should be 'caused' by changes in food production infrastructure.

## Data and methods

All data and code to conduct our analysis are freely available [74]. To conduct our analysis, we first estimated changes in population density, then we integrated estimates of population density with human bone isotope data and modeled paleoclimate data.

To estimate changes in population, we used a database of archaeological radiocarbon from CWA. We then used the R package rcarbon [75] to construct Kernel Density Estimates (KDEs) to estimate changes in the population of the three areas in CWA over the last 3,700 years cal BP. In all areas, we used ShlCal2020 to calibrate the radiocarbon ages [76]. We start our analysis at 3,700 cal BP because, around this time, we observe a more continuous human occupation of the region [24, 50, 77]. We construct mean KDEs in an attempt to help control for the biases of sampling intensity, preservation, and the non-linear radiocarbon calibration curve [8, 14–16, 25, 75, 78]. Specifically, we construct mean KDEs in each area of CWA by running 200 simulations with a bandwidth of 50, and we control for the oversampling of

particular archaeological sites by clustering dates by site using the h-function (h = 100) in rcarbon. We then calculate the mean of the 200 KDEs and sum the mean KDE into 30 year bins. These procedures smooth the KDE to capture the long-term trend over time and reduce intragenerational fluctuations over shorter time-scales induced by calibration and/or biases introduced by site oversampling. Further, we construct the KDEs from 4,000 to 200 cal BP and then trim the sequences to 3,700 to 300 cal BP. We trim the sequences to avoid edge effects (i.e., a lack of radiocarbon dates on material remains prior to 4,000 and, especially, after 300 cal BP).

In all areas, we are conservative and do not use a global taphonomic adjustment (e.g., [79]) in our main analysis for three reasons (see S1 Appendix for more details). (1) After 6,000 cal BP, the global taphonomic adjustment displays considerable uncertainty in the parameters that determine the shape of the archaeological context loss function over time [79], across geomorphic contexts [80], and does not take into account the converse taphonomic process that larger sites and greater landscape modification by humans may preserve more material for archaeological sampling. (2) The mathematical procedure suggested for the global taphonomic adjustment introduces complexity and uncertainty into the data by amplifying noise and, potentially, real signals of oscillations in occupational intensity [1]. (3) In the S1 Appendix, we illustrate that using the global taphonomic adjustment does not change the long-term structure of the KDEs used in our main analysis, and we document that the density of material remains in a stratified cave deposit tracks changes in the mean KDEs constructed in our main analysis [50]. Thus, the long-term expansion of occupation intensity in the region does not appear to simply result from the loss of archaeological contexts over time.

We evaluate expectations #1–3 in three steps. First, we analysed population density–*per capita* fitness 'phase plots' to detect potential signatures of demographic transitions and count the number of successive potential demographic transitions in each area. These plots allow us to compare the structure of population expansion in the three areas and assess expectation #1. To construct these plots, we calculated the *per capita* growth rates of the mean KDEs as: ln $(MKDE_{t+1}/MKDE_t)$. Please note that the mean *per capita* growth rate is an estimate of *per capita* fitness because they are equivalent. However, we use the term *per capita* growth rate here to acknowledge that we cannot directly observe the fitness of an average individual over time in archaeological cases and that the KDEs contain signals of changes in population, economic complexity, and noise [22].

Once we calculated the *per capita* growth of the mean KDEs in each area, we plotted the mean KDEs (an estimate of population density) against the *per capita* growth rate of each area in order of time from the oldest 30 year bin to the youngest. This allows us to observe how estimates of population density and *per capita* growth jointly change over time. As noted in the Model dynamics section, an inverse shaped V or U in these plots indicates a potential demographic transition. A hook shaped curve may signal a recession to equilibrium, and, in real systems, the trajectory of population change may display a cycle, which manifests as a circular to oval curve traveling around a relatively constant mean population density over time.

Second, to evaluate expectation #2 we fit a logistic model to capture the trend over time in the mean KDEs, and we checked for an association between the consumption of energy dense resources and the relative size of the mean KDE. We use the logistic model because the ISM proposes that population expansion over thousands of years may follow innovations that, at first raise a population's limit a lot. However, innovations then display diminishing returns, leading to a slowing of increases in a population's limit until a region hits a temporary innovation ceiling.

As a first approach, to estimate changes in the production and consumption of energy dense resources, we use stable isotopes on human bone. Specifically, we synthesize data on stable isotopes from human bones from previously published studies [6, 7, 57, 59], and we

organize the isotope data by the temporal phases discussed above and listed in Table 1. We use changes in $\delta^{13}C$ carbonate, collagen, and $\delta^{15}N$ collagen to estimate changes in the extraction and consumption of carbohydrate rich plants. Carbon from carbonate tissue provides an estimate of the whole diet, while carbon isotopic values from collagen tracks the consumption of protein, primarily derived from animals. As such, carbonate $\delta^{13}C$ includes carbon from both plants and animals. For example, among omnivores a linear relationship exists between $\delta^{13}C$ carbon values in collagen and carbonate, but only within a $C_3$ or a $C_4/marine$ protein group [81, 82]. Nitrogen isotopes potentially track the trophic position of consumers in ecosystems, with more consumption of animal protein increasing $\delta^{15}N$ values in bone collagen. In CWA, the main $C_4$ resource available was maize and the main wild plants available were $C_3$ resources [6, 7, 57, 59]. Thus, we can use stable isotopes from human bone to potentially track changes in the consumption of maize and wild, $C_3$ plants. In our main analysis, we focus on changes in $\delta^{13}C$ carbonate because these values track relative changes in the whole diet, and we provide more details on all isotopes in the S1 Appendix.

Finally, to evaluate expectation #3, we use causal impact analysis [83] to assess the effect of long-term climate change relative to innovations in food production on population density. To estimate the suitability of past climate for human populations, we use PaleoView [84] to estimate mean annual precipitation in the three areas. PaleoView couples outputs from the TRaCE21ka experiment [85–87], a Community Climate System Model, version 3 (CCSM3), and a global coupled atmosphere-ocean-sea ice-land general circulation model (AOGCM) with 3.75 degree latitude-longitude resolution on land. PaleoView re-grids the climate data to provide a 2.5 x 2.5 degree resolution on a global scale from 20,050 BC to 1989 AD. We use the grid unit 32.5 to 35 degrees south and 70 to 67.5 degrees west to estimate rainfall in the Northern and Central areas. We use the grid cell 35 to 37.5 degrees south and 70 to 67.5 degrees west to estimate rainfall in the Southern area. We used PaleoView because it provides comparable paleoclimate estimates of rainfall between the areas and because paleoclimate data are scarce in CWA. To estimate net primary productivity, we generated data sets that estimate rainfall every 30 years over the past 4,000 years. We focus on rainfall because CWA is a dryland environment that was so dry during the Mid-Holocene that hunter-gatherers virtually abandoned the region. However, after 4,000 cal BP, occupation intensity increased in CWA and productivity was controlled largely by the availability of water in the region.

Causal impact analysis divides a time-series of interest (mean KDE estimate of population density) and covariates (rainfall) up into a pre-treatment and a post-treatment data set. The pre-treatment data set is used to model the effect of rainfall on mean KDE values. In this case, we use the identification of periods of demographic transition identified in step 1 above to divide the time-series into pre-treatment and post-treatment data sets. The causal impact package uses a locally weighted Bayesian linear model. The model takes into account the local trend, covariates, and error [83 e.g., Fig 2]. The model is then used to predict the population density (mean KDE) of the post treatment data set.

For example, the pre-treatment data set for the Northern area occurs from the initial occupation of the area at 3,700 cal BP until the spread of domesticates sometime between 3,000 and 2,400 cal BP and evidence of three successive demographic transitions beginning at 2,950 cal BP. The spread of domesticates is an innovation or "intervention" that may change the underlying behavior of a time-series relative to past controllers (e.g., rainfall). In this case, we can model the effect of rainfall on population density over the pre-intervention period. This allows us to ask a counterfactual question: What would population density have been, given rainfall, if there had been no innovation in the production of food? The difference between the observed and modeled population density during the post-intervention period is the causal impact of innovations on changes in population density. If the difference between the observed

density and modeled population density (mean KDE) is low, then we can say that potential innovations had little impact relative to the long-term climate controlled productivity of resources modeled during the pre-intervention period. This assumes that rainfall does not depend on innovations in human infrastructure systems and that rainfall continually impacts the productivity of resources, regardless of the infrastructure for producing food.

## Archaeological results

Our analysis indicates three main results. First, partly consistent with expectation #1, the longest period of population expansion occurs in the Northern area, and shorter periods of population expansion occur in the Central and Southern areas. Only in the Northern area do we observe potentially successive demographic transitions. Second, consistent with expectation #2, the Northern and Central areas display population expansions that associate with a greater consumption of energy dense, plant resources. In the southern area, changes in population density also associate with changes in the consumption of resources that may have included the consumption of more energy dense plants and wild animals. Third, consistent with expectation #3, following suspected demographic transitions, population density increases more than one would expect based on the relationship between mean climate and population density in the preceding generations.

### Expectation #1: Population expansion

Fig 5 compares the population dynamics of the Northern, Central, and Southern areas. These plots document how the mean KDE (estimated population density) and the *per capita* growth rate of the mean KDE (estimated *per capita* fitness) change in tandem over time. The arrows on each plot illustrate the direction of change in the two variables over time, and the red dots estimate the central value of the mean KDE when the two variables display one or more cycles (i.e., the curve travels in a circular shape around the red dot). As noted above, the plots in Fig 5 allow us to detect potential demographic transitions with the distinctive inverse V (or U) shape noted in Fig 1 of the Model dynamics section, and the plots reveal two main patterns.

First, we only observe sustained demographic transitions in the Northern area. To observe this pattern, read Fig 5A by starting your gaze at the point labeled 'I' with a KDE value very near zero and *per capita* growth just below 0.3. This point is the beginning of the time-series. Now, let your gaze follow the arrows along the curve as they trace the mutual change in the mean KDE and *per capita* growth over time. At first, the system displays a cycle (i.e., a circular shape). However, after the mean KDE passes the point labeled 'D', the system displays successive inverse U shaped curves over time that are the signature of demographic transitions. The pattern of sustained waves of KDE expansion occurs until the mean KDE reaches a value of approximately 0.011. At this point, the system displays a single cycle, and then another potential demographic transition to a cycle with a higher mean KDE. Conversely, the Central and Southern areas only display one small demographic transition between cycles and one larger demographic transition. In short, consistent with expectation #1, the Northern area displays more consistent population expansion, and the Central and Southern areas both display less consistent population expansion and spend more time oscillating around a relatively stable mean KDE value.

Second, we note here that the mean KDEs are in arbitrary units. Thus, one should not compare the absolute values of the plots. The plots in Fig 5 illustrate the structure of population dynamics between the three areas (i.e., the shapes of the curves on each plot). In this regard, there is a key difference between these plots and the plots that we generated in our analysis of the ISM. The ISM, under the parameters that we study, only generates partial cycles as the

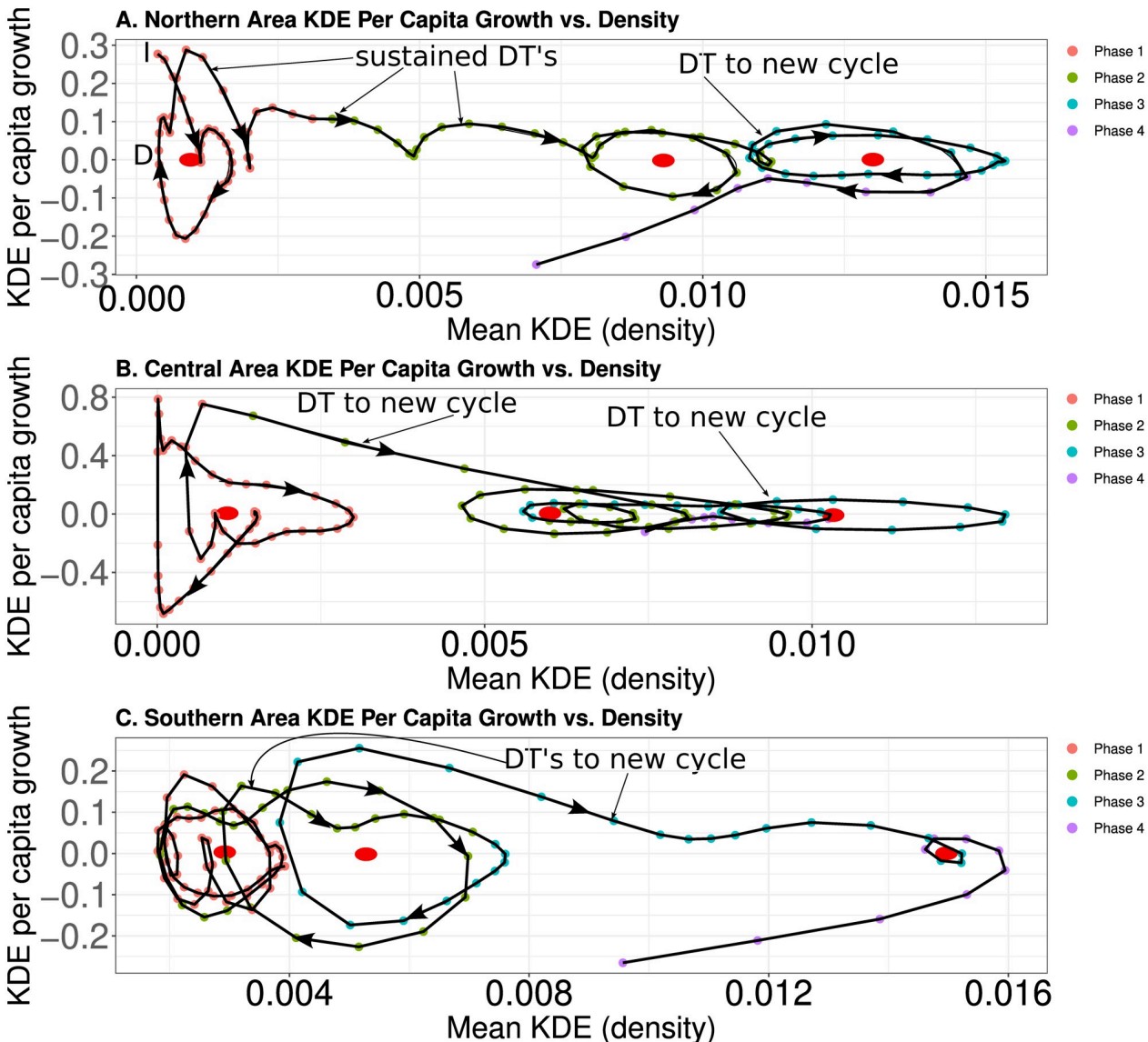

**Fig 5. A-Plot of mean KDE (x-axis) against the *per capita* growth rate of the mean KDE (y-axis) in the Northern area.** Data points connect in order from oldest to youngest. The arrows indicate the direction of change over time. Red dots estimate the central KDE value when one or more cycle is detected. B-Plot of mean KDE (x-axis) against the *per capita* growth rate of the mean KDE (y-axis) in the Central area. C-Plot of mean KDE (x-axis) against the *per capita* growth rate of the mean KDE (y-axis) in the Southern area.

population overshoots and then declines to equilibrium. In the real data, we observe cycles of overshoot, decline, and then overshoot again, if a demographic transition does not occur. Such cycles are more prevalent in the Central and Southern regions and indicate that these population systems never entered a simple, stable equilibrium.

## Expectations #2: Population expansion and energy dense resource consumption

Fig 6A and 6B document the prolonged population expansion in the Northern area over time. The blue dashed boxes on Fig 6B highlight the suspected demographic transitions documented

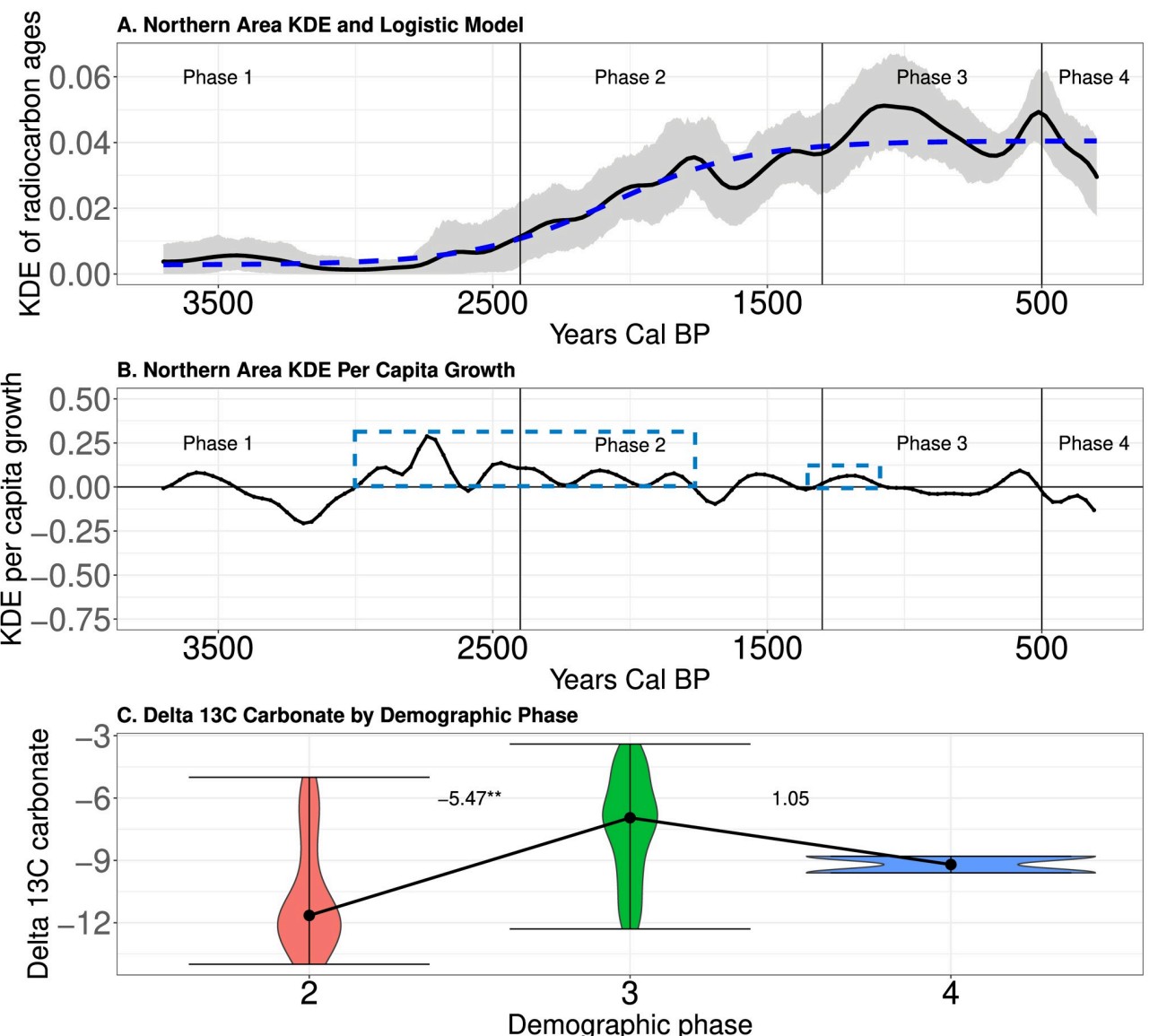

**Fig 6. A-Mean KDE (black curve) and 95% confidence envelope of simulated KDEs (grey shading) in the Northern area.** B-The *per capita* growth rate of the mean KDE over time. The blue dashed boxes indicate periods of repeated demographic transitions. C-Violin plots of $\delta^{13}C$ carbonate by phases. Black dots indicate the medians of the distributions and lines illustrate the change in median between phases. Numbers indicate the value of a pairwise Conover test. ** indicates p<0.05.

above in Fig 5A. We observe a population expansion that begins about 2,950 cal BP and lasts for 990 years (i.e., 33, 30 year generations)! The first secure dates on domesticated plants occur during this time [59, 60]. The second potential demographic transition between population cycles is highlighted in Fig 6B between Phases 2 and 3 beginning at about 1,410 cal BP. Fig 6C illustrates that changes in resource consumption associate with population expansion (Kruskal–Wallis chi-squared = 22.51, df = 2, p-value <0.05), with $\delta^{13}C$ carbonate values increasing from Phase 2 to Phase 3. This suggests that the consumption of $C_4$ resources was significantly different among phases, with more maize likely consumed during Phase 3 [7]. In short, as population density expanded, people shifted their diet to the production and consumption of more $C_4$ resources, likely maize.

Fig 7A and 7B document the population dynamics identified in the Central area over time. As noted in Fig 5B, this area experienced two potential demographic transitions separated by long periods of the mean KDE oscillating around a relatively constant value. For instance, population density increased abruptly with the adoption of domesticates [55, 71, 77, 88] around 2,400 cal BP, and the population then oscillated around a new, higher population density until approximately 1,210 cal BP, when population density again expanded. Fig 7C illustrates that, just as in the Northern area, people became more dependent upon $C_4$ resources over time. Fig 7C documents a trend from Phase 1 to 2 to 3 of increasing $\delta^{13}C$ carbonate values (Kruskal–Wallis chi-squared = 18.07.13, df = 3, p-value <0.05). This indicates that an increased

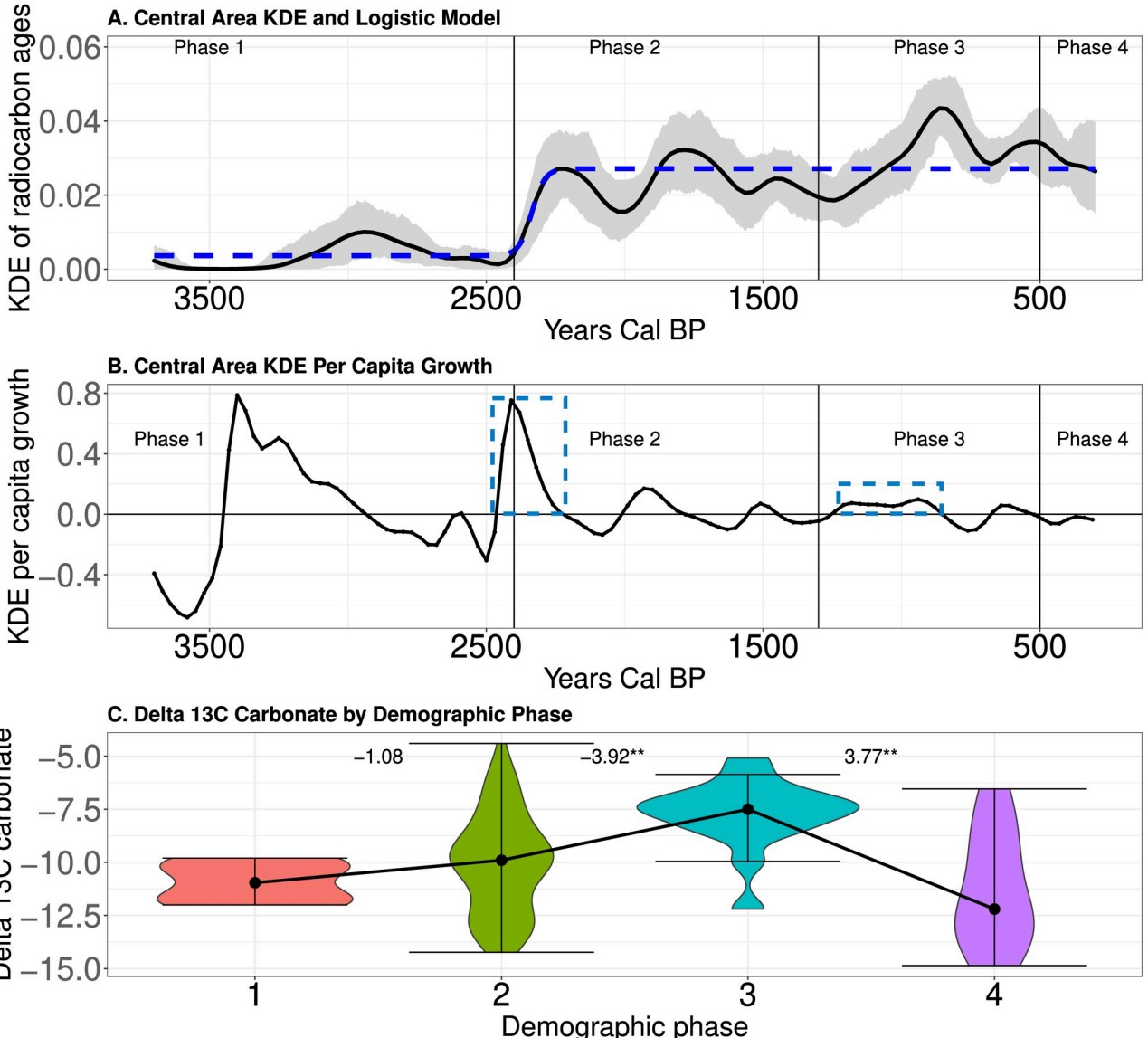

**Fig 7. A-Mean KDE (black curve) and 95% confidence envelope of simulated KDEs (grey shading) in the Central area.** B-The *per capita* growth rate of the mean KDE over time in the Central area. The blue dashed boxes indicate periods of potential demographic transitions. C-Violin plots of $\delta^{13}C$ carbonate by phases. Black dots indicate the medians of the distributions and lines illustrate the change in median between phases. Numbers indicate the value of a pairwise Conover test. ** indicates p<0.05.

production and consumption of $C_4$ resources, like maize, associates with the two periods of increased estimates of population density.

Fig 8A and 8B document the population dynamics identified in the southern area over time. The blue boxes on Fig 8B highlight potential demographic transitions identified in Fig 5C during Phase 2 around 2,030 cal BP and in Phase 3 beginning at about 1,000 cal BP. Fig 8C illustrates that, coincident with the increase in mean KDE during Phases 2 and 3, we observe a trend of decreasing $\delta^{13}C$ carbonate values (Kruskal–Wallis chi-squared = 10.37, df = 3, p-value <0.05). This is the exact opposite of the pattern noted in the Northern and Central areas, and potentially indicates more exploitation of $C_3$ resources, including energy dense wild plant resources.

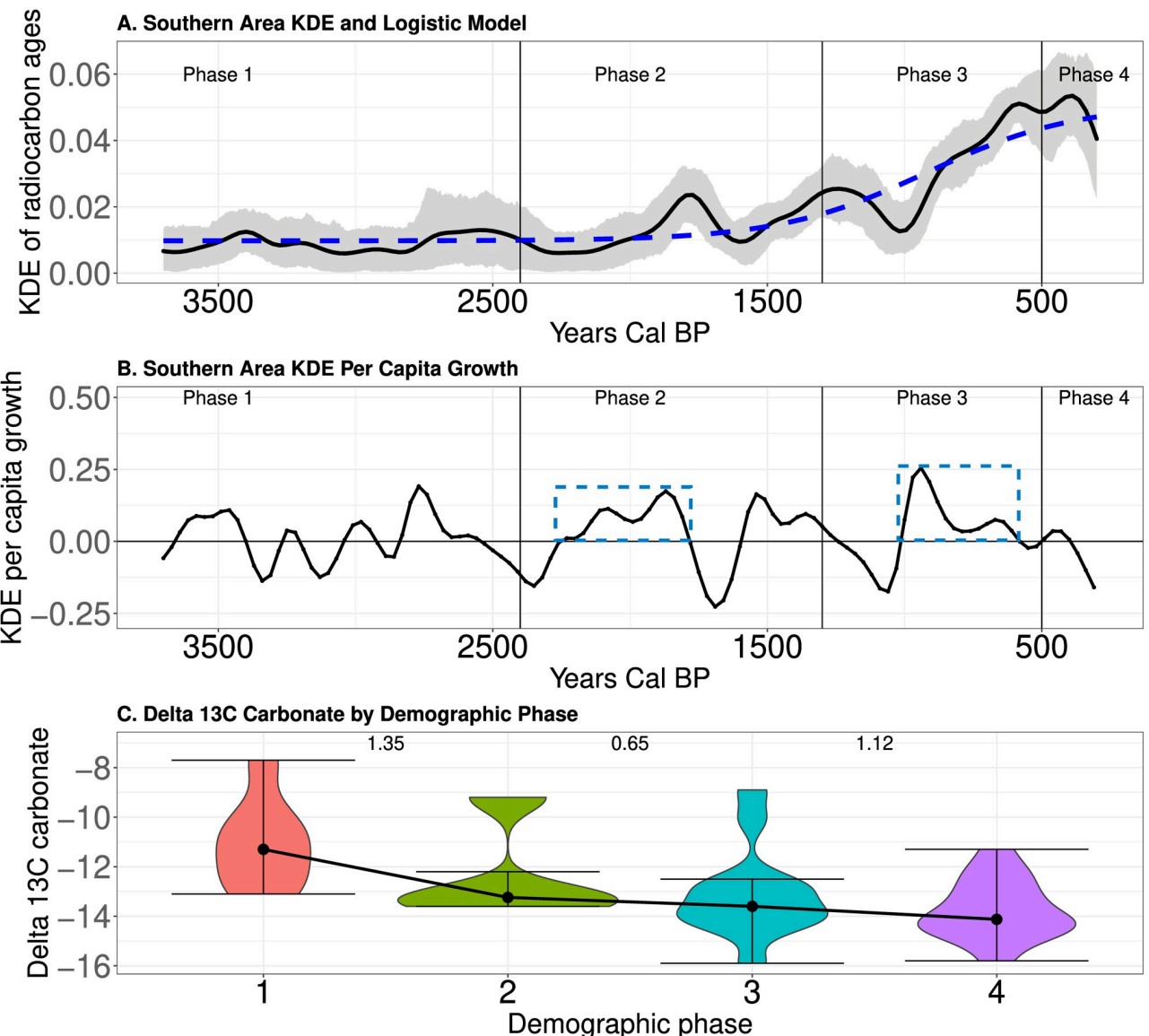

**Fig 8. A-Mean KDE (black curve) and 95% confidence envelope of simulated KDEs (grey shading) in the Southern area.** B-The *per capita* growth rate of the mean KDE over time in the South. The blue dashed boxes indicate periods of potential demographic transitions. C-Violin plots of $\delta^{13}C$ carbonate by phases. Black dots indicate the medians of the distributions and lines illustrate the change in median between phases. Numbers indicate the value of a pairwise Conover test. ** indicates p<0.05.

In sum, in all three areas we observe shifts in the consumption of resources associated with increases in estimates of population density over time. We observe an increase in $\delta^{13}C$ carbonate values during periods of higher estimated population density in the regions where populations adopted maize (Northern and Central areas) and a decrease in $\delta^{13}C$ carbonate values where populations did not adopt maize (Southern area). Based on these results and our S1 Appendix, we suggest that these shifts were toward more carbohydrate production and consumption from plants during the phases of higher estimated population density (see Discussion below). This is consistent with expectation #2 that population expansion associates with evidence for the production and consumption of more energy dense resources.

### Expectation #3: Long-term mean climate and population expansion

Fig 9 illustrates the causal impact analysis of long-term changes in rainfall and food production on the initial demographic transitions suspected in each area. For example, in Fig 9A, the green curve illustrates the modeled effect of rainfall on the mean KDE (grey shaded area is a 95% prediction interval). The statistical model trains on data between 3,700 and 2,950 cal BP. The extension of the green curve from 2,949 to 1,410 cal BP (the relevant period of successive demographic transitions) illustrates the expected effect of rainfall on the mean KDE in the absence of the adoption of domesticates. The distance between the black curve and the green curve between 2,949 to 1,410 cal BP captures the potential pointwise causal effect of the adoption of domesticates on the mean KDE (graphed in Fig 9B). In all three areas, the pointwise distance between the expected and actual KDE increases markedly following the "intervention" of suspected innovations in food production (Fig 9B, 9D and 9F).

Specifically, in the Northern area (Fig 9A and 9B) the KDE is 560% (C.I., 467%, 677%, Posterior tail-area probability: 0.001) higher in the interval of 2,949 to 1,410 cal BP than we would expect given the relationship between rainfall and the KDE modeled from 3,700 to 2,950 cal BP. In the Central area (Fig 9C and 9D), the observed KDE is 566% (C.I., 397%, 828%, posterior tail-area probability: 0.001) higher in the interval of 2,499 to 1,210 cal BP than we would expect. Finally, in the Southern area (Fig 9E and 9F), the observed population density is 95% (C.I., 80%, 115%, posterior tail-area probability: 0.001) higher in the interval of 2,030 to 1,000 cal BP than we would expect. In short, following the adoption of subsistence economies based on the more intensive use of plants, estimated population density increased in each area more than one would expect if rainfall's long-term impact on ecosystem productivity controlled population density alone.

Fig 10 illustrates the causal impact analysis of long-term changes in rainfall and food production on the the second period of demographic transitions suspected in each area. As above, in all three areas, we observe increases in population density above what we would expect based on rainfall alone. In the North, the observed KDE is 36% (C.I. 30%, 42%, posterior tail-area probability: 0.001) higher in the interval of 1,320 to 500 cal BP than we would expect given the relationship between rainfall and the KDE modeled from 1,930 to 1,321 cal BP. In the Central area, the observed KDE is 30% (C.I. 22%, 40%, posterior tail-area probability: 0.001) higher in the interval of 1,210 to 500 cal BP than we would expect. In the South, the observed KDE is 135% (C.I. 111%, 164%, posterior tail-area probability: 0.0013) higher in the interval of 1,000 to 500 cal BP than we would expect.

## Discussion and conclusion

Recent decades have witnessed a revolution in the study of long-term human population ecology, documenting three main patterns. (1) Humans sometimes experience exponential-like expansion over thousands of years during the Holocene. (2) Over hundreds of years to a few

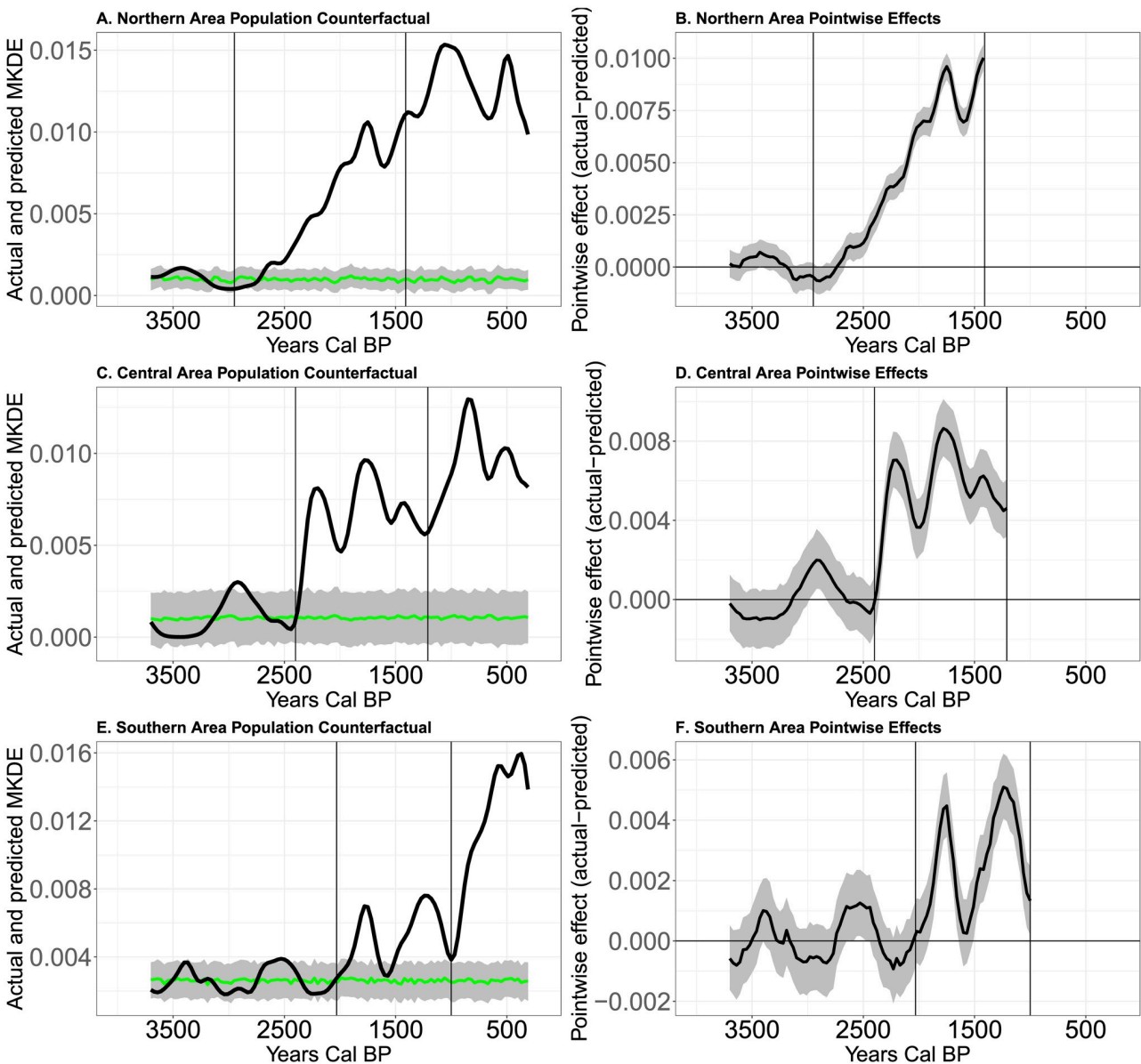

**Fig 9. A-Modeled (green curve) and actual mean KDE values over time in the Northern area.** Grey shaded area is the confidence envelope of predicted mean KDE values based on modeled rainfall. B-Pointwise causal effect of suspected subsistence/social innovation on the mean KDE in the Northern area. C-Modeled (green curve) and actual mean KDE values over time in the Central area. Grey shaded area is the confidence envelope of predicted mean KDE values based on modeled rainfall. D- Pointwise causal effect of suspected subsistence innovation on the mean KDE in the Central area. E-Modeled (green curve) and actual mean KDE values over time in the Southern area. Grey shaded area is the confidence envelope of predicted mean KDE values based on modeled rainfall. F-Pointwise causal effect of suspected subsistence innovation on the mean KDE in the Southern area.

millennia, human populations sometimes experience recessions and oscillations. (3) Human populations display a demographic transition associated with the adoption of agriculture and, potentially, repeated waves of demographic transitions among both agricultural and hunter-gatherer archaeological regions associated with innovations in food production. These basic patterns suggest three fundamental questions: (1) What mechanisms enable and constrain the long-term, exponential-like expansion of human populations? (2) Why do some regions

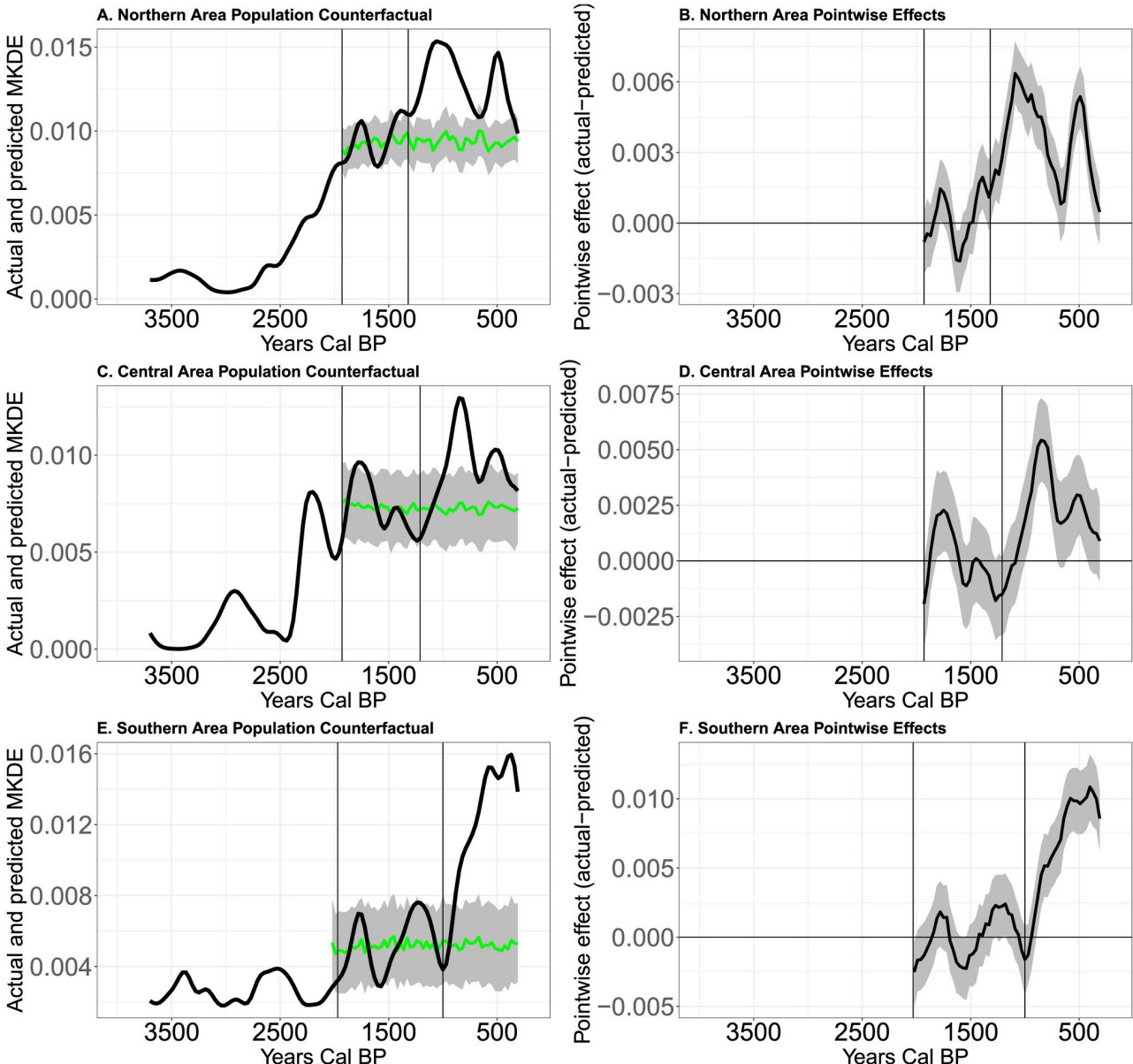

**Fig 10. A-Modeled (green curve) and actual mean KDE values over time in the Northern area.** Grey shaded area is the confidence envelope of predicted mean KDE values based on modeled rainfall. B- Pointwise causal effect of suspected subsistence innovation on the mean KDE in the Northern area. C-Modeled (green curve) and actual mean KDE values over time in the Central area. Grey shaded area is the confidence envelope of predicted mean KDE values based on modeled rainfall. D- Pointwise causal effect of suspected subsistence innovation on the mean KDE in the Central area. E-Modeled (green curve) and actual mean KDE values over time in the Southern area. Grey shaded area is the confidence envelope of predicted mean KDE values based on modeled rainfall. F-Pointwise causal effect of suspected subsistence innovation on the mean KDE in the Southern area.

display more inter-generational stability than other regions? (3) Is the ADT a revolutionary event, or is the ADT one manifestation of a more general process in which innovations in the production of resources sometimes generate demographic transitions among human societies? Answering these questions requires a two pronged research effort that develops models useful for generating hypotheses about the diversity of population growth trajectories observed among archaeological cases [1, 2, 13, 26], and a library of case studies that integrate multiple

lines of evidence to evaluate hypotheses and expectations reasoned from formal models. In this paper, we contributed to this effort by building on and studying an Ideal Specialization Model (ISM). We then provided a concrete example of how the model can be used to investigate the long-term expansion of population in Central West Argentina (CWA). Our research contributes to understanding the diversity of long-term trajectories of population expansion displayed among archaeological regions and generates questions for future research.

## Long-term population expansion and the ADT

The ISM illustrates that the innovation of infrastructure for producing food and the mean climate of a region can both impact the long-term expansion of human populations (Fig 1). In the ISM, generating population expansion over thousands of years requires that the mean climate must change such that the productivity of ecosystems continually improves, or populations in a region must continually open and fill new resource spaces through social and technological innovations. The two processes can work in concert, counteract each other, or one process (e.g., innovation) may overpower the other (e.g., climate change). Thus, to understand the processes that control the long-term expansion of human populations, one strategy is to compare regions with similar climates that we suspect, from previous archaeological research, display different trajectories of social-technological change. CWA provides one such location.

Our analysis of the archaeological patterns from CWA suggests that innovations in the production of food generated population expansion, and the ADT can be considered a specific manifestation of this general mechanism. For example, the Northern area of CWA experienced longer and more robust population expansion than the Central and Southern areas. However, the Central and Southern areas each display brief periods of population expansion. In all three cases, the expansion of population associates with changes in food production, maize and domesticates in the Northern and Central areas and wild plants in the Southern area (Figs 6–8). Further, population expansion is larger than we would expect from long-term climate trends alone in all three areas (Figs 9 and 10). Consistent with the ISM, this evidence suggests a sequential adoption of more intensive food production, though the process was not progressive and could be reversed [1, 7].

An important question to answer is why the Northern area experienced successive waves of demographic transitions following the adoption of domesticates while the other areas experienced long periods of population oscillation around a relatively constant mean (Fig 5). Three observations suggest some processes that may explain these different trajectories of population expansion in the three areas.

First, we currently have no evidence of a 1,000 year trend of increasing climate suitability in the Northern area. In fact, the Northern and Central areas experienced similar climate changes and no sustained increase in rainfall that might solely account for population expansion (Figs 9 and 10). Second, the Central region displays two abrupt periods of population expansion separated by periods of population cycles. The first expansion occurs with the spread of maize into the region. The second occurred at the beginning of Phase 3, coincident with evidence for more consumption of maize in the region. Third, our analysis of $\delta^{13}C$ carbonate suggests that the consumption of maize was similar in the Northern and Central areas. Yet, the Central area displays fewer ceramics, no evidence of pit structures and village communities, less evidence of painted ceramics, and later evidence of occupation across all elevations. This hints that innovations in social organization and settlement integration may have been important to sustain a population and innovation ratchet after the adoption of agriculture in the Northern area. Indeed, recently Boone and Alsgaard [89] propose that the

adoption of agriculture is not critical per se to generating a population and innovation ratchet. Rather, the ability of a population to organize labor and scale-up cooperation is the key limit, and raising this limit depends on generating surplus value associated with social signaling. We speculate that if changes in the production of resources are unaccompanied by additional innovations in collective action, the surplus value of social signaling, and the ability to modify a landscape, then the adoption of domesticates, in and of itself, simply leads to a brief burst of population expansion and then long periods of relative stasis around a demographic, social-technological cycle.

Moving forward, at least two lines of additional research are required to continue evaluating the mechanisms that drive long-term population expansion in CWA. (1) Human bone stable isotopes are potentially subject to multiple interpretations and are complicated in the region by nonlinear distributions of $C_3$/$C_4$ plant resources [90]. The evidence potentially suggests that diets changed over time in association with changes in population density, as we would expect, but more work is needed. For instance, the sample size in the Southern area is small, and previous work indicates that variation in human bone isotope values is linked to spatial differences in the isotope ecology of ecosystems in the region and beyond [91, 92]; thus, large samples of individuals over time, controlling for ecosystem, are better for detecting changes in resource consumption over time [5]. More importantly, the argument that innovations in food production drive population expansion would strengthen if we could develop higher resolution time-series of changes in cooking and processing technology in the regions to complement the human stable isotope data sets. In the Southern area, there is evidence that people began to more intensively use plants such as *Prosopis sp.* during the Late Holocene [93], and developing time-series of such resource use would allow us to better evaluate the link between population expansion and the use of carbohydrate rich plants.

(2) The causal impact analysis of the effects of mean rainfall on estimated population density suggests that population density increased more than we would expect due to long-term climate trends alone in all three areas. More work is needed on this front. There are few paleoclimate data from our study area, and the paleoclimate model that we used could have errors that impact our results. Paleoclimate time-series for the region would help resolve the uncertainty associated with such potential errors. Further, as documented in the analysis of the ISM, climate variation operates at multiple scales. It is unclear whether an internal social dynamic may have led to the adoption of more energy dense plant resources evidenced in the human bone isotope data or the higher frequency climate variability mechanism discussed as part of the ISM (Fig 3). The temporal resolution of the archaeological record is not fine grained enough yet in CWA to explore these alternatives.

(3) One could argue that the Northern area was simply more suitable in terms of climate, the geomorphology of streams, etc. for maize production than the Central and Southern areas. For example, from North to South in CWA, the region transitions from a rainfall regime dominated by summer rainfall to one dominated by winter rainfall. This is potentially important because winter rainfall associates with higher upfront infrastructure costs to produce maize, which needs moisture during flowering and grain filling during summer months [57, 94]. Thus, these biophysical conditions may have set the stage for more reliable innovations associated with the production of maize in the Northern area and, thus, more consistent population expansion. To test this idea, we need a larger comparison of the adoption of maize agriculture and population dynamics on the eastern and western slopes of the Andes. While summer rainfall occurs on the eastern side of the Andes, it does not on the West. Thus, the latitude range of our study area provides a sort of natural experiment to investigate if maize production and population dynamics follow different trajectories of change in winter vs. summer rainfall environments.

## Population stability and innovation

Moving forward, the ISM, like the models of previous researchers [33, 34, 36, 41], encodes the Boserupian idea that population pressure drives innovations. There is a minimum tolerable well-being greater than demographic replacement (i.e., *per capita* growth = 0) that, once crossed, motivates the adoption of innovations in the production of food. In the ISM, this assumption can create a dynamic in which *per capita* growth remains positive for dozens of generations as populations continually ratchet up their production of food. However, two factors explored in the ISM can arrest the continual waves of demographic transitions. (1) If a system encounters an innovation ceiling (a reduction in latent innovations), then populations cannot expand in a Boserupian fashion. As illustrated in Fig 2, under conditions in which humans have a delayed impact on resources, approaching an innovation ceiling produces short-term population instability in the form of a recession of carrying capacity. (2) Constraints on the ability of populations to detect signals of population pressure also snuff out the potential for a population ratchet and populations end up in equilibrium (Fig 3). However, stochastic climate variability may help populations innovate despite being near a limit (Fig 3), but more work is needed to understand the operation of this mechanism.

Although we did not study differences in population stability between the areas of CWA explicitly, we would like to note a direction for future research. In the Central and Southern areas, *per capita* growth rates often oscillate around a long-term limit, displaying evidence of persistent population cycles. A potential question is why some regions might experience more dramatic cycles than other regions. The ISM is an equilibrium based model, though population dynamics can display damped oscillations under some parameter combinations. It may be that the social-political dynamics related to the spread of innovations and constraints on the ability of a population to detect and respond to signals of population pressure are important to investigate to understand why some regions display persistent cycles and why some cycles are more dramatic than others [26].

For example, our results potentially mirror recent results from North America in which large innovations in food production associate with longer periods of population expansion but also larger recessions–the Adaptive Capacity Tradeoff Hypothesis [1]. Consistent with this hypothesis, the Northern area displays more evidence of prolonged population expansion, but also a 15 generation (450 year) population recession during Phase 3 after peak population density and the population system entered into a cycle. Conversely, the Central and Southern areas do not display repeated innovations nor prolonged expansions and recessions. These regions display shorter, but persistent cycles of growth and recession around a relatively constant mean population density. More work is needed to investigate whether larger innovations in food production consistently associate with more severe population recessions and, potentially, more dramatic population cycles.

## Conclusion

The Ideal Specialization Model and our analysis contribute to understanding the long-term expansion of human populations. Most researchers agree that population dynamics are controlled by a complex interaction between climate change, changes in human made infrastructure, and population density. Yet, the nature of the interactions, when they will lead to repeated innovations and the long-term expansion of population, or when they will lead to population recessions, or long-term population cycles remain challenging questions. Our work builds on previous studies and helps shed light on the above questions. In particular, the adoption of domesticated plants does not automatically lead to long-term population expansion. Rather, the adoption of domesticates, likely, must be accompanied by additional changes

in infrastructure to produce greater surpluses of crops and/or reduce the social costs of transporting and accessing resources. If not, then populations may remain quite consistent in the sense that they oscillate around a long-term, infrastructure controlled limit, constantly buffeted by climate variability much more than populations consistently experiencing an innovation ratchet.

## Supporting information

**S1 Appendix. Supporting information appendix on taphonomic loss and stable isotopes.** (PDF)

## Acknowledgments

We would like to thank two anonymous reviewers for their time and effort critiquing the manuscript. Their efforts have greatly improved the the manuscript. We would also like to thank the Museo Moyan, Museo FFyL Canals Frau, Museo de Historia Natural de San Rafaeal-Humberto Lagiglia, and Museo Regjonal de Malargue-Jorge Luna for support accessing samples and data.

## Author Contributions

**Conceptualization:** Jacob Freeman, Adolfo F. Gil, Eva A. Peralta, Fernando Franchetti, José Manuel López, Gustavo Neme.

**Data curation:** Adolfo F. Gil, Eva A. Peralta, Fernando Franchetti, José Manuel López, Gustavo Neme.

**Formal analysis:** Jacob Freeman.

**Funding acquisition:** Adolfo F. Gil.

**Investigation:** Jacob Freeman.

**Methodology:** Jacob Freeman, Eva A. Peralta, Fernando Franchetti, José Manuel López, Gustavo Neme.

**Resources:** Adolfo F. Gil, Fernando Franchetti, José Manuel López, Gustavo Neme.

**Writing – original draft:** Jacob Freeman, Adolfo F. Gil, Eva A. Peralta, Fernando Franchetti, José Manuel López, Gustavo Neme.

**Writing – review & editing:** Jacob Freeman, Adolfo F. Gil, Eva A. Peralta, Fernando Franchetti, José Manuel López, Gustavo Neme.

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
