## [Decision Letter · Decision Letter 0]

1 Apr 2024

PONE-D-24-04698Deep time population expansions and recessions in Central West ArgentinaPLOS ONE

Dear Dr. Freeman,

Thank you for submitting your manuscript to PLOS ONE. After careful consideration, we feel that it has merit but does not fully meet PLOS ONE’s publication criteria as it currently stands. Therefore, we invite you to submit a revised version of the manuscript that addresses the points raised during the review process.

We look forward to receiving your revised manuscript.

Kind regards,

Radu Iovita

Academic Editor

PLOS ONE

Journal Requirements:

"We are thankful for the support of research grant PICT 2021-I-A-00891"

"We are grateful for support for Past Global Changes (PAGES) project, which in turn received support from the Swiss Academy of Sciences and the Chinese Academy of Sciences. We are greatful for the support of research grant PICT 2021-I-A-00891, and to the Museo Moyan, Museo FFyL Canals Frau, Museo de Historia Natural de San

Rafaeal-Humberto Lagiglia, and Museo Regjonal de Malargue-Jorge Luna for support accessing samples and data."

"We are thankful for the support of research grant PICT 2021-I-A-00891"

5. We note that Figure 4 in your submission contain map images which may be copyrighted. All PLOS content is published under the Creative Commons Attribution License (CC BY 4.0), which means that the manuscript, images, and Supporting Information files will be freely available online, and any third party is permitted to access, download, copy, distribute, and use these materials in any way, even commercially, with proper attribution. For these reasons, we cannot publish previously copyrighted maps or satellite images created using proprietary data, such as Google software (Google Maps, Street View, and Earth). For more information, see our copyright guidelines: http://journals.plos.org/plosone/s/licenses-and-copyright.

A. You may seek permission from the original copyright holder of Figure 4 to publish the content specifically under the CC BY 4.0 license.  

B. If you are unable to obtain permission from the original copyright holder to publish these figures under the CC BY 4.0 license or if the copyright holder’s requirements are incompatible with the CC BY 4.0 license, please either i) remove the figure or ii) supply a replacement figure that complies with the CC BY 4.0 license. Please check copyright information on all replacement figures and update the figure caption with source information. If applicable, please specify in the figure caption text when a figure is similar but not identical to the original image and is therefore for illustrative purposes only.

Additional Editor Comments:

**Please respond to Reviewer 1's concerns with the methodology and clarity of the manuscript. Additionally, please upload code that works, this is essential for reproducibility.**

Reviewers' comments:

Reviewer's Responses to Questions

**Comments to the Author**

1. Is the manuscript technically sound, and do the data support the conclusions?

Reviewer #1: Partly

Reviewer #2: Yes

2. Has the statistical analysis been performed appropriately and rigorously? 

Reviewer #1: Yes

Reviewer #2: Yes

3. Have the authors made all data underlying the findings in their manuscript fully available?

Reviewer #1: No

Reviewer #2: Yes

4. Is the manuscript presented in an intelligible fashion and written in standard English?

Reviewer #1: Yes

Reviewer #2: Yes

5. Review Comments to the Author

Reviewer #1: In this article, the authors present an analysis of summed probability distributions of radiocarbon dates from Central Western Argentina (CWA) to demonstrate multiple population expansions and regressions during the Holocene in this region. These demographic changes are then interpreted in the context of the Ideal Specialization Model to argue for successive population expansions that are associated with multiple innovations in food production instead of a single agricultural demographic transition.

While this paper does attempt to make good use of available data to answer questions about demographic trends in the Holocene record of CWA, there are major issues with this paper including: 1) lack of taphonomic correction for radiocarbon dates and 2) lack of concrete archaeological data for contextualizing the authors' claims. For these reasons, I would suggest major revisions for this article.

Please find my full comments in the attached Word document.

Reviewer #2: While I am not familiar with the simulation models the authors use, I am aware of the data on which they are based and their previous publications in which they have been discussed. Therefore I consider that this work fits perfectly with the Plos One criteria.

The manuscript seems to me to be an excellent contribution to the understanding of the topic discussed about the population pulses in different sectors of the Central West of Argentina, in relation to the total or partial adoption of corn within their diet.

My only objection is that there is a need to delve into socio-political aspects related to the interaction between different populations of the macro region of study that may have influenced changes in the subsistence patterns analyzed.

6. PLOS authors have the option to publish the peer review history of their article (what does this mean?). If published, this will include your full peer review and any attached files.

Reviewer #1: No

Reviewer #2: **Yes: **Mónica Alejandra Berón

---

## [Author Response · Author response to Decision Letter 0]

2 Jun 2024

Provided in the pdf of the cover letter and response letter.

---

## [Decision Letter · Decision Letter 1]

2 Jul 2024

PONE-D-24-04698R1A model of long-term population growth with an application to Central West ArgentinaPLOS ONE

Dear Dr. Freeman,

Thank you for submitting your manuscript to PLOS ONE. After careful consideration, we feel that it has merit but does not fully meet PLOS ONE’s publication criteria as it currently stands. Therefore, we invite you to submit a revised version of the manuscript that addresses the points raised during the review process.

**Hi, please see minor revisions requested by the reviewer, which seem very reasonable to me.**

We look forward to receiving your revised manuscript.

Kind regards,

Radu Iovita

Academic Editor

PLOS ONE

Journal Requirements:

Reviewers' comments:

Reviewer's Responses to Questions

**Comments to the Author**

1. If the authors have adequately addressed your comments raised in a previous round of review and you feel that this manuscript is now acceptable for publication, you may indicate that here to bypass the “Comments to the Author” section, enter your conflict of interest statement in the “Confidential to Editor” section, and submit your "Accept" recommendation.

Reviewer #1: (No Response)

2. Is the manuscript technically sound, and do the data support the conclusions?

Reviewer #1: Yes

3. Has the statistical analysis been performed appropriately and rigorously? 

Reviewer #1: Yes

4. Have the authors made all data underlying the findings in their manuscript fully available?

Reviewer #1: Yes

5. Is the manuscript presented in an intelligible fashion and written in standard English?

Reviewer #1: Yes

6. Review Comments to the Author

Reviewer #1: The updated version of this paper is much improved. Specifically, I greatly appreciate that the authors took the time to test the effects of taphonomic corrections on their radiocarbon dates and to explain their reasons for using uncorrected curves. This decision is now much better justified. I also appreciate the updates to the code and that the authors now provide all data necessary to successfully run the scripts. Furthermore, the inclusion of more contextualizing information about the archaeology of the CWA and the reorganization of how this case study is presented makes it easier to follow the logical flow of the paper. The new Figure 5 is a great addition to this paper; it helps tie together the description of the ISM and the archaeology. Ultimately, I think there are just a few additional minor revisions the authors could make to their “Model dynamics” section to improve its comprehensibility and and to support the conclusions of this paper even better.

1) In Figure 1, the numbered curves are confusing because they seem to suggest 4 infrastructure systems despite the authors saying they assume a ceiling of A2. Does the occurrence of an innovation necessarily push a population into a new infrastructure system or does the model allow for multiple innovations to occur in A1 and A2?

2) It is only in the caption of Figure 2 that it is explained that A = A2 – A1. This should be moved to the main text (perhaps on line 248) to make it clearer that A = 3 does not mean A3 exists.

3) It is extremely difficult to see the black curve in Figure 3 with the overlaid green and red lines. This makes it very hard for the reader to comprehend the effects of climate variability to keep the population from reaching equilibrium or to force oscillation between infrastructure systems (i.e. the patterns that authors describe in lines 293-295). Is there a better way to visualize this, especially since the green curve is not meant to be a central tendency of the black curve but a separate pattern altogether?

Minor comments:

There is a spelling error in Figure 1 – should be “population density”

In figure 1 caption, should “s” be capitalized to match S in the previous equations?

In figure 3b, is it possible to color the lines to match the green used in figure 3a that is showing what will occur without climate variability?

7. PLOS authors have the option to publish the peer review history of their article (what does this mean?). If published, this will include your full peer review and any attached files.

Reviewer #1: No

---

## [Editor Report · Decision Letter 2]

10 Jul 2024

A model of long-term population growth with an application to Central West Argentina

PONE-D-24-04698R2

Dear Dr. Freeman,

We’re pleased to inform you that your manuscript has been judged scientifically suitable for publication and will be formally accepted for publication once it meets all outstanding technical requirements.

Kind regards,

Radu Iovita

Academic Editor

PLOS ONE
---

## [Editor Report · Acceptance letter]

15 Jul 2024

PONE-D-24-04698R2 

PLOS ONE

Dear Dr. Freeman, 

I'm pleased to inform you that your manuscript has been deemed suitable for publication in PLOS ONE. Congratulations! Your manuscript is now being handed over to our production team.

Kind regards, 

on behalf of

Dr. Radu Iovita 

Academic Editor

PLOS ONE